# Application of In Vivo MRI Imaging to Track a Coated Capsule and Its Disintegration in the Gastrointestinal Tract in Human Volunteers

**DOI:** 10.3390/pharmaceutics14020270

**Published:** 2022-01-24

**Authors:** Sarah Sulaiman, Pavel Gershkovich, Caroline L. Hoad, Matthew Calladine, Robin C. Spiller, Snow Stolnik, Luca Marciani

**Affiliations:** 1Nottingham Digestive Diseases Centre, National Institute for Health Research (NIHR), Nottingham Biomedical Research Centre, Nottingham University Hospitals NHS Trust, University of Nottingham, Nottingham NG7 2UH, UK; sarah.sulaiman@nottingham.ac.uk (S.S.); robin.spiller@nottingham.ac.uk (R.C.S.); 2School of Pharmacy, University of Nottingham, Nottingham NG7 2QL, UK; pavel.gershkovich@nottingham.ac.uk (P.G.); pazmc@exmail.nottingham.ac.uk (M.C.); snow.stolnik@nottingham.ac.uk (S.S.); 3Sir Peter Mansfield Imaging Centre, School of Physics and Astronomy, University of Nottingham, Nottingham NG7 2QX, UK; Caroline.L.Hoad@nottingham.ac.uk

**Keywords:** targeted drug delivery, delayed-release formulations, magnetic resonance imaging, fat imaging, large intestine

## Abstract

Oral specially coated formulations have the potential to improve treatment outcomes of a range of diseases in distal intestinal tract whilst limiting systemic drug absorption and adverse effects. Their development is challenging, partly because of limited knowledge of the physiological and pathological distal gastrointestinal factors, including colonic chyme fluid distribution and motor function. Recently, non-invasive techniques such as magnetic resonance imaging (MRI) have started to provide novel important insights. In this feasibility study, we formulated a coated capsule consisting of a hydroxypropyl methylcellulose (HPMC) shell, coated with a synthetic polymer based on polymethacrylate-based copolymer (Eudragit^®^) that can withstand the upper gastrointestinal tract conditions. The capsule was filled with olive oil as MRI-visible marker fluid. This allowed us to test the ability of MRI to track such a coated capsule in the gastrointestinal tract and to assess whether it is possible to image its loss of integrity by exploiting the ability of MRI to image fat and water separately and in combination. Ten healthy participants were administered capsules with varying amounts of coating and underwent MRI imaging of the gastrointestinal tract at 45 min intervals. The results indicate that it is feasible to track the capsules present in the gastrointestinal tract at different locations, as they were detected in all 10 participants. By the 360 min endpoint of the study, in nine participants the capsules were imaged in the small bowel, in eight participants in the terminal ileum, and in four in the colon. Loss of capsule integrity was observed in eight participants, occurring predominantly in distal intestinal regions. The data indicate that the described approach could be applied to assess performance of oral formulations in undisturbed distal gastrointestinal regions, without the need for ionizing radiation or contrast agents.

## 1. Introduction

The aim of targeted oral formulations, including delayed-release dosage forms, is to deliver the active agent at the intended site in the gastrointestinal tract and in that way regulate the absorption rate, the plasma concentration-time profile, and the pharmacodynamics of the formulated drug [1].

Despite the importance of colon-targeted formulations, the knowledge of formulation transit and release is still limited. Whilst knowledge of formulation behavior in the upper gastrointestinal (GI) tract is improving, the lower GI tract remains much less explored, partly because of the difficulty of accessing the anatomical areas under physiological (unprepared) conditions [2]. Much of the available information derives from gamma scintigraphy studies conducted initially in the 1980s and 1990s. Whilst pioneering and very informative, nuclear medicine methods have limited spatial resolution and provide limited anatomical or functional information on the surrounding organs [3,4,5]. Other innovative methods such as magnetic marker monitoring (MMM) have been used. These techniques use a magnetic detector to follow the fate of solid formulations along the GI tract [6,7,8,9,10,11], providing good temporal resolution but again limited anatomical information.

Imaging techniques such as magnetic resonance imaging (MRI) have the potential to enrich current knowledge by providing new non-invasive and non-ionizing insights on the undisturbed gastrointestinal (GI) tract, with the real-time assessment of the physiological environment surrounding drug products, including water content, chyme properties, and mixing by the intestinal wall contractions [12,13,14]. MRI has been gaining increasing attention in the pharmaceutical field to inform understanding of the environment that drug products will be experiencing and the development of pharmaceutical products [14,15,16,17,18,19,20,21,22].

One of the first studies, conducted by Schiller et al., applied MRI to investigate the simultaneous passage of non-disintegrating oral formulations and quantify the fluid presence in the intestines in the fasted and fed state. The study concluded that fluid presence remains mainly unchanged in the colon in both fasted and fed states and introduced the concept of gastrointestinal tract fluid pockets, which increase in numbers after food administration. The study was able to locate the oral formulation tested (capsules 16.8 mm in length and 4.6 mm in diameter) and link movement of these non-disintegrating gel capsules to food intake [17]. Sager at al. took advantage of the superparamagnetic properties of incorporated iron oxide to visualize the transfer of immediate-release hard gelatin capsules in the GI tract [23]. Grimm et al. encapsulated dried pineapple and used MRI to monitor the behavior of acid-resistant capsules in vivo, exploiting the MRI contrast agent properties of the fruit on T1-weighted water imaging [24]. MRI has also been used in other studies to image formulations, but primarily in the upper GI tract, including non-disintegrating oral formulations, release of MRI contrast agents from capsules, multi-particulate surrogates and gadolinium-labelled liposomes [24,25,26,27,28,29,30,31,32].

Building on this work, our study aimed to test the hypothesis that it would be possible to use MRI to track the location of fat-filled oral coated capsules throughout the GI tract and to image their loss of integrity by taking advantage of the ability of MRI to image fat and water separately and in combination. We report here the use of enteric coated oral capsules and the feasibility of their in vivo MRI imaging following oral administration to 10 healthy human participants.

## 2. Materials and Methods

### 2.1. Capsule Development

Size 0 capsules (21 mm in length by 7 mm in diameter), made from hydroxypropyl methylcellulose (HPMC, Your Supplements, Stockport, UK) without coloring agent were manually filled with 0.65 mL olive oil (Tesco Stores Ltd., Welwyn Garden City, Hertfordshire, UK) as the imaging-visible agent. The capsules were then sealed with Methocel^TM^ K4M glue and manually dip-coated. A 20:1 *w*/*w* stock mixture of ethanol and water was prepared, with the ratio chosen according to the manufacturer’s protocol. The Eudragit^®^ S 100 polymer powder (kindly gifted by Evonik Operations GmbH, Essen, Germany) was added into an aliquot of the solvent mixture of ethanol and water while stirring, until the polymer was completely dissolved. Triethyl citrate (plasticizer) was added into a separate, equal amount of the ethanol and water solvent mixture while stirring. The plasticizer solution was then slowly poured into the Eudragit^®^ solution while stirring. This coating solution was then stored at room temperature for 24 h to cool down and be ready for use the next day.

Each HPMC-filled capsule was weighed first and then dip-coated manually with Eudragit^®^ S 100 coating solution in consecutive dipping (coating) cycles. Following the coating procedure, the coated capsules were visually inspected for visible unevenness of the coating; only those capsules with smooth surface appearance were used in the study and their data on mass gain measurements reported. When the coating procedure was completed, each capsule was stored individually to avoid tacking and allowed to dry overnight. Weight gain was monitored throughout the coating process and at 24 h.

A range of concentrations of Eudragit^®^ S 100 coating solution and increasing number of dipping cycles were chosen based on literature data and preliminary in vitro observations. Coating solution concentrations of 2.3%, 4.6%, 6.7%, 8.8%, 10.7% and 13.0% of Eudragit^®^ S 100 were prepared in order to investigate dependence of capsule weight gain on the coating polymer solution concentration. Weight gain was measured throughout the manual dip-coating in triplicate to quantify the amount of coating placed on each capsule. The surface area of a capsule was calculated using the spherocylinder area formula, knowing the length and diameter of the capsule.

### 2.2. In Vitro Disintegration Study

The disintegration behavior of the uncoated and coated capsules was tested in the USP compatible basket apparatus (ZT 120 light, ERWEKA GmbH, Langen, Germany). In this protocol, 700 mL of 0.1 M hydrochloric acid was used for the ‘acid stage’, simulating the gastric fluid, and the capsules with different coating weight (Table 1) were exposed for 1 h. The capsules were then immersed into 700 mL of phosphate buffer pH 6.8, termed here the ‘buffer stage’, simulating the intestinal fluids. The experiments were performed at 37 ± 2 °C and a stirring speed of 30 ± 1 strokes/min as per the United States Pharmacopeia (USP) [33,34]. Triplicates of uncoated and capsules coated by 1, 2 or 3 dipping cycles in 10.7% Eudragit^®^ S 100 coating solution were tested. Time to loss of capsule integrity, as well as to complete disintegration of capsules were recorded.

### 2.3. In Vivo Study Design and Participants

The experiment was an open-label, one arm, feasibility study. The study protocol was approved by the University of Nottingham’s Faculty of Medicine and Health Sciences Research Ethics Committee, Nottingham; approval number 402-1910. All participants gave written informed consent, and had no contraindications to MRI.

All 10 participants (4 male and 6 female) were healthy volunteers aged 22–41 years with no history of underlying cardiac or gastrointestinal disorders or symptoms. All participants had a normal-range body mass index (BMI) between 20.5 and 24.4 kg/m^2^, and reported no food intolerances or allergies.

The participants were asked to fast from 8:00 pm the evening prior to the MRI study day. They were asked to skip breakfast and to report to the imaging facilities fasted. Prior to the study process, all volunteers underwent an initial fasted baseline MRI scan to check that the stomach was fasted, determine anatomy and to plan upcoming scans. Following that, the participants were administered a coated capsule with 240 mL of water in a standing position. A range of coated capsules were prepared and administered to participants, as summarized in Table 2.

Each MRI scan of a participant lasted approximately 15 min. Participants were scanned at predetermined time intervals, with the images acquired every 45 min post-administration of the capsule. Image data were acquired until the investigators were confident that the capsule could no longer be detected for loss of filling or up to 8 h post ingestion. When the coated capsule exited the stomach, a meal was administered that consisted of a cheddar and ham sandwich, a 25 g pack of salted crisps, a butter croissant, and 500 mL of water (total energy content 670 Kcal).

At each time point, for each participant, the gastrointestinal tract location of the capsule was noted from the image data. Additionally, the capsule’s intact appearance or loss of integrity, intended as apparent deformation of the capsules and/or release of some or all the oil filling, was noted. When the capsule presence in the image stacks could not be detected, this was labelled NO or Not Observed.

### 2.4. MRI Acquisition

The participants were scanned in a supine position in a 3 Tesla Philips Achieva MRI scanner (Philips, Best, The Netherlands). They lay supine on the scanner table with a 16-channel receiver placed around the abdomen. A multiple-echo, mDIXON [35] sequence was used with echo time 1 = 1.32 ms, echo time 2 = 2.2 ms, flip angle = 20°, repetition time = 10 ms, field of view = 250 × 350 mm^3^ and acquired resolution = 1.8 × 1.8 × 4.4 mm^3^. Coronal views of the abdomen were acquired, divided into short breath hold stacks. The reconstructed mDIXON sequence yielded 4 image types (water, fat, water and fat in phase, water and fat out of phase). At each time point, the MRI procedure took approximately 15 min including set up, scout imaging and calibrations.

In 4 participants, the time taken from the stomach to the appearance in the colon was known, and therefore, a small bowel transit time could be calculated from time of appearance in the colon minus time of last detection in the stomach.

The data are presented as mean ± standard deviation (SD).

## 3. Results

### 3.1. Capsule Coating

A representative example of the coated capsules ready for use is shown in Figure 1A. Figure 1B summarizes the weight gained by the HPMC capsules upon coating with the solution of Eudragit^®^ S 100 at different concentrations. The weight gain for the 10 capsules manufactured for the human study is reported in Table 2, both in terms of absolute weight gain and estimated weight gain (mass) per capsule unit surface area at 24 h after preparation.

The amount of coating deposited on the capsules increased with increased concentrations of Eudragit^®^ S 100 solution and with the number of dipping (consecutive coating) cycles. Weight gain variability also increased with higher polymer concentrations and number of dipping cycles.

### 3.2. Capsule Disintegration

The data from the disintegration tests in acid (stomach) and buffer (intestinal) conditions are reported in Table 1. In the disintegration tests (in triplicates), the uncoated capsules lost their integrity and released their first drop of oil in 6.2 ± 2.2 min in the simulated stomach (acid) stage. The uncoated capsules had disintegrated entirely within 11.5 ± 2.9 min in the acid stage. All of the coated capsules remained intact in the acid stage. The capsules manufactured using the lowest amount of coating (9.2 mg) released their first drop of oil in 15.5 ± 3.2 min during the intestinal (buffer) stage. The capsules manufactured using 18.2 mg of coating lost their shape and released the first drop of oil after 116.9 ± 20.4 min in the buffer stage, whereas the capsules manufactured using the higher 25.9 mg coating maintained their integrity in the buffer stage for over 6 h, when the disintegration experiment was stopped, with no signs of loss of shape or oil release.

### 3.3. Capsule Imaging Studies In Vivo

All 10 participants swallowed the capsules easily and tolerated the serial MRI scanning procedures and the relatively long study day well.

Figure 2 shows an example of MRI tracking of an intact capsule inside the body of one participant, from the stomach through the small bowel until the capsule is seen as completely empty at the end of the distal ascending colon, having released the oil ‘payload’. In this case, the release of the oil happened in between the imaging time points and was not captured. By contrast, Figure 3 shows a partially filled and deformed capsule at the hepatic flexure region of colon in a different participant 180 min post-administration. Figure 3A displays the whole body section with the anatomy (the fat and water out of phase image plane) to pinpoint the location of the capsule. Figure 3B displays the fat only imaging mode and depicts not only the partial filling remaining in the capsule, but also the oil released from the capsule, present in the colon chyme and close to the colon wall. Similar results obtained from another participant are shown in Figure 4. This time the capsule was imaged at the bottom of the ascending colon 135–180 min post-administration, with some oil filling released and distributed along the colon wall.

Table 2 summarizes details of the capsule detection in all 10 participants. All capsules were detected intact in the stomach at t = 45 min after ingestion. After this time, the capsule’s transit through the GI tract varied across participants. In nine participants, the capsules were detected in the small intestine. Only in one participant, the capsule was imaged losing integrity in the stomach. This was the capsule with the lowest weigh gain of 0.02 mg/mm^2^. All other capsules reached the small intestine apparently intact. Eight capsules reached the terminal ileum and four the colon. Loss of integrity was imaged directly in eight of the capsules at various locations as detailed in Table 2.

From the timing of the images, it was also possible to calculate small bowel transit time as the time taken from the stomach to the appearance in the colon, which was 169 ± 22 min (*n* = 4, mean ± SD).

## 4. Discussion

This is, to the best of our knowledge, the first study to assess the feasibility of using MRI to locate an oil-filled, gastro-resistant (enteric) polymer-coated capsule throughout the gastrointestinal tract and evaluate its integrity. Olive oil was chosen as an MRI-visible marker fluid filling for the capsules for two principal reasons. The first one is that typically conventional MRI scanners have the unique ability to capture and image signal from fat separately from the water signal and to produce different algebraic combinations of fat and water in phase or out of phase from each other. The olive oil contained in the coated capsules provides high MRI fat signal and virtually no MRI water signal, thus allowing us to image the intact capsule as well as oil released from a capsule that has lost its integrity.

Here, as illustrated in the MRI images in Figure 2, Figure 3 and Figure 4, the fat and water out of phase imaging combination was able to clearly visualize the capsule, whilst providing good anatomical detail, enabling assessment of the capsule’s spatial location within the gastrointestinal tract. Furthermore, considering imaging of oil-containing formulations in the distal intestinal tract, one would not expect to find much fluid fat in the chyme in the colon, and, therefore, the signal from the oil in the capsule should be clearly visible. The second reason is that olive oil is an inexpensive, safe, food-grade material, and there are no ethical issues arising from the capsules disintegrating and releasing oil in the bowel of healthy human participants.

The design of the gastro-resistant, polymer-coated capsule used in this study, the choice of specific Eudragit^®^ polymer, and the range of concentrations of the coating solution were based on the literature [36,37,38,39,40,41,42,43,44,45,46,47,48,49,50]. The selection of HPMC as the capsule shell took into consideration that HPMC is often used as a pre-coating material in gastro-resistant/enteric-coated formulations, as gelatin capsules do not appear to have good adhesion properties when directly coated with Eudragit^®^ polymers [38,47]. In addition, HPMC properties seem to favor the gradual drug release in the colon [43].

The disintegration test undertaken on Eudragit^®^ S 100 coated HPMC capsules illustrates dependence of the capsules’ disintegration on weight gain due to coating. Uncoated capsules disintegrated in acidic environment (simulated stomach conditions), whilst all tested coated capsules remained intact for the duration of the acidic environment test (1 h). In pH 6.8 buffer, simulating intestinal conditions, capsules with 9.2 mg of coating material lost their integrity to release the encapsulated oil, and then disintegrated entirely in less than 20 min. On the other hand, the capsules with 18.2 and 25.9 mg of coating layer maintained their integrity in the intestinal buffer stage for approximately 2 and 6 h, respectively, until the release of oil was observed, indicating that the intestinal capsule integrity is proportional to the coating weight gain. These in vitro findings corroborate the in vivo MRI imaging findings. More specifically, the capsule with 9.2 mg weight gain (administered to participant number 1) showed loss of integrity while still present in the stomach. In the participants that were administered capsules with the weight gain of 18.2 and higher, the capsules were emptied from the stomach intact and travelled further down the gastrointestinal tract. The MRI data obtained initially guided iteratively the selection of coating, so that the performance of the capsule with 18.2 mg coating during in vitro as well as in vivo studies meant that these were predominantly used in the study. In three participants, we tested capsules with much higher coating gain (36.0 and 52.5 mg) and resistance to disintegration, the latter shown in the in vitro disintegration test, and observed loss of their integrity in two participants, with intact capsule observed in the transverse colon region of one participant.

It should be noted that coating of a surface (e.g., capsule shell) with a polymeric material may result in the creation of an uneven layer, particularly if adhesion forces between two materials are low, as illustrated above for gelatin capsule shells, and this was the reason this study used HPMC capsules. We visually inspected all capsules tested and used in the MRI study for any unevenness and selected only those with even, smooth coating. In further studies and optimizations, one would aim to measure the layer thickness and its uniformity at the capsule shell. Furthermore, future studies could evaluate the bioavailability of active pharmaceutical compounds incorporated within the capsule.

One limitation of this study was the relatively prolonged interval between consecutive imaging time points. The participants were scanned every 45 min and, once the imaging was completed, asked to sit upright in an adjacent room until the following scan time point. This imaging protocol was adopted to make the study day more comfortable for the participants. However, in this way, events that may have occurred between scans could not be recorded.

In Table 2, when the capsule presence in the image stacks could not be detected, this was labeled as ‘Not Observed’. In most cases, this occurred after the capsule at the previous time point was observed to be losing its shape and integrity, or release of encapsulated oil was noticed (text in *italic* in Table 2), thus losing the capsule’s imaging signal. In some cases (for example, in participant number 5) this might also have meant that at the time of data acquisition, physiological motion could have blurred the capsule signal, effectively masking the visual identification of the capsule’s presence. The three-dimensional image acquisition scheme used in this study can be particularly sensitive to motion, and one could in future deploy two-dimensional, multi-slice acquisition sequences to minimize the effect of motion on the images.

The study indicates that Eudragit S 100 coatings higher than 0.02 mg/mm^2^ of polymer would be needed to ensure capsule gastro-resistance in vivo, based on the data for participant number 1 versus those from participants administered capsules with coatings equal to or higher than 0.04 mg/mm^2^. The latter coating resulted in capsules that, in all but two of the participants, reached, and were imaged in, more distal gastrointestinal regions. In this study, we were able to image the presence of coated capsules in the colon in three participants, as illustrated in Figure 3 and Figure 4, and showed direct imaging of colon delivery and spread of the oil ‘payload’ inside the colon.

This type of non-invasive imaging study could assist in understanding the distribution of formulations and pharmacologically active ingredients in the colon. Increased knowledge of colon-targeted formulation location and disintegration timing could also inform advanced in silico approaches.

## 5. Conclusions

This feasibility study demonstrates that a combination of MRI imaging and fat-filled, coated capsules can extend the reach of current MRI methods to image more distally along the bowel. This can provide information on the formulation transit, as well as information on the release of the payload in the colon. This method could be used in the future to assess performance of delayed-release formulations in an undisturbed gastrointestinal environment, without using ionizing radiation. This approach could also be used to deliver active pharmaceutical ingredients whilst monitoring the transit and actual arrival in the colon. Further work to optimize the imaging timings and protocol is needed, as well as demonstration of the use of the capsule in conjunction with an active drug payload.

## Figures and Tables

**Figure 1 pharmaceutics-14-00270-f001:**
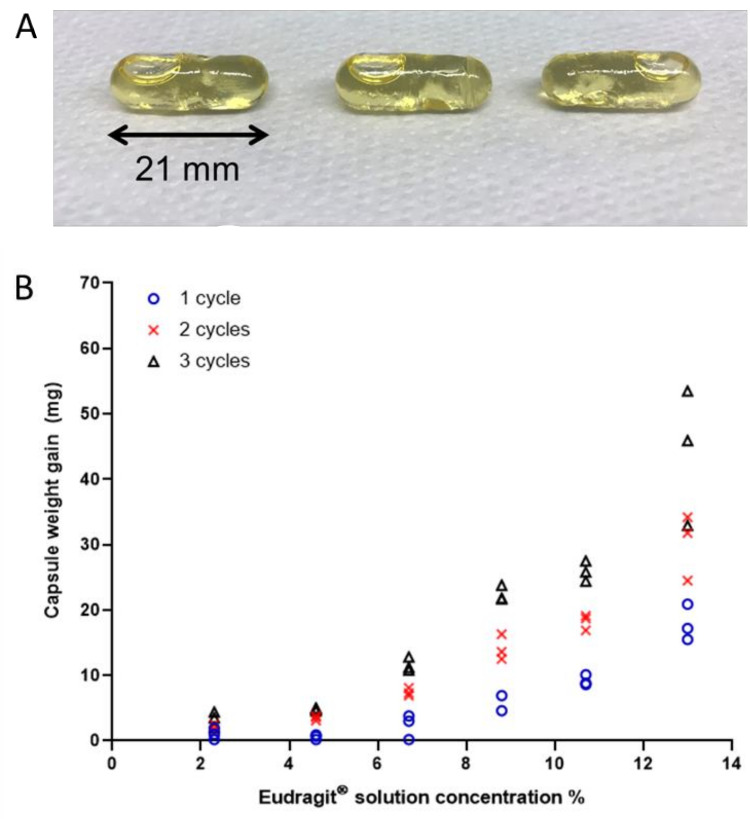
(**A**) A representative example of HPMC capsules filled with olive oil and coated with Eudragit^®^ S 100. The capsule measures 21 mm length by 7 mm diameter. (**B**) Capsule weight gain measured in triplicate at 24 h after one, two and three dipping (coating) cycles for different concentrations of Eudragit^®^ S 100 solution.

**Figure 2 pharmaceutics-14-00270-f002:**
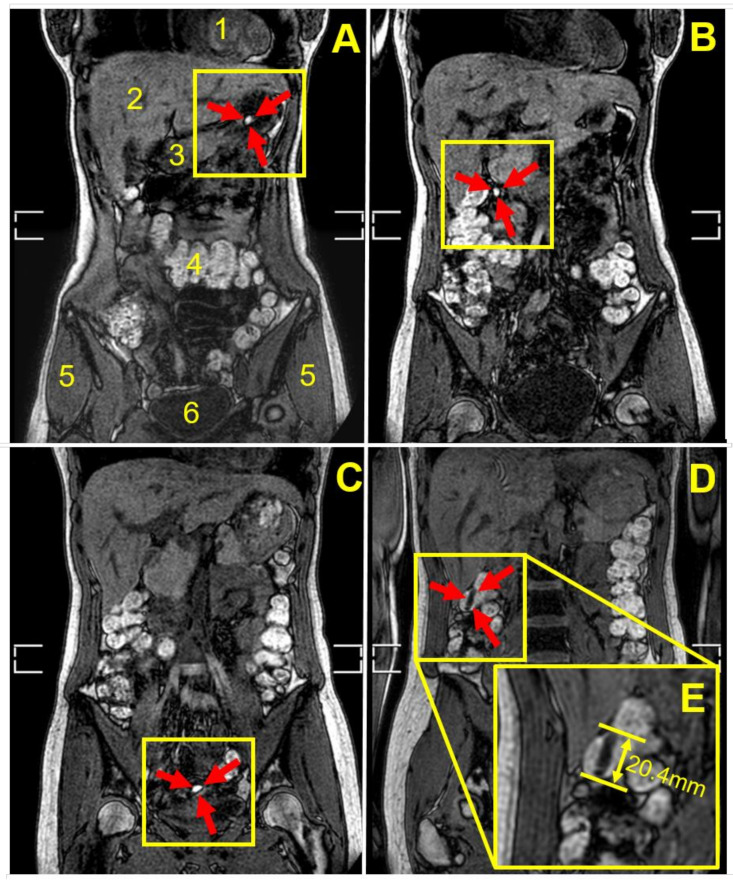
Coronal, fat and water out of phase MRI images of the abdomen of participant number 2 after consumption of the coated capsule co-administered with 240 mL of water. Anatomical landmarks are indicated in (**A**) by numbered labels as heart (**1**), liver (**2**), stomach antrum (**3**), transverse colon (**4**), gluteus medius muscle (**5**) and bladder (**6**). The white brackets at the side of the body derive from the fusion of the two separate breath-hold image stacks acquired. The filled capsule appears bright. It was located intact and floating in the body of the stomach 45 min after consumption as shown in panel (**A**) indicated by the red arrows inside the yellow box. The capsule was detected later in the small bowel in the duodenum 90 min (**B**) and ileum (**C**) 135 min after consumption. Finally, the capsule was located in the large intestine in the ascending colon, near the hepatic flexure, 180 min after consumption (**D**). By this time the capsule appeared to be empty of oil but still with an almost complete capsule shape, visible in (**D**) as a dark spherocylindrical object that measured 20.4 mm in length (inset (**E**)). The white brackets at the side of the body derive from the fusion of the two separate breath-hold image stacks acquired.

**Figure 3 pharmaceutics-14-00270-f003:**
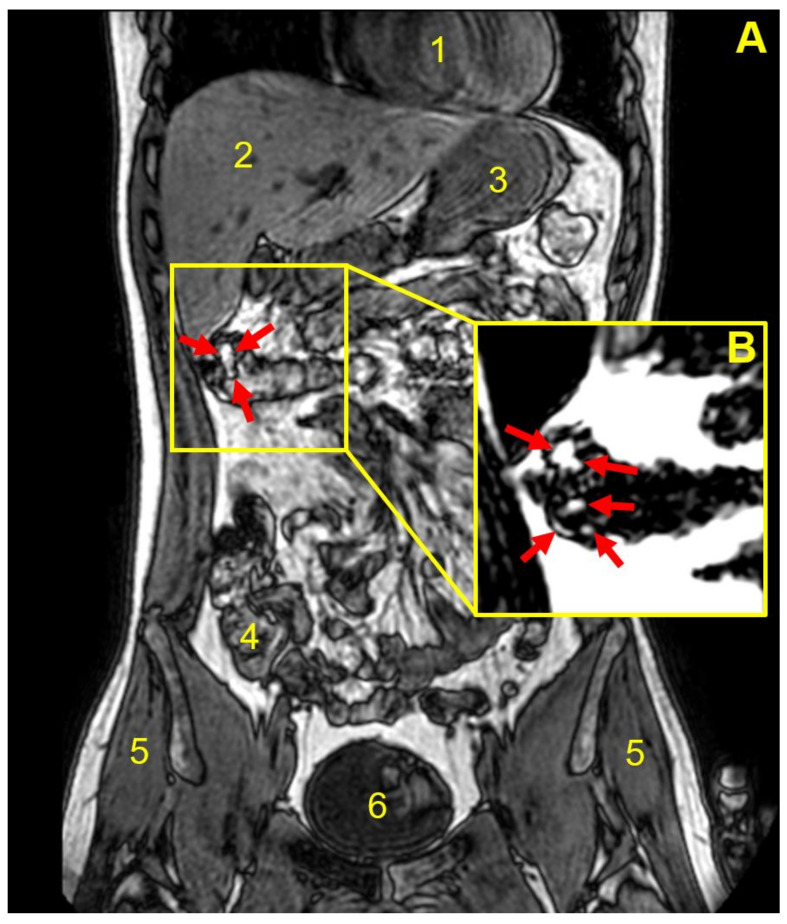
Coronal, fat and water out of phase MRI image of the abdomen of participant number 3, acquired 180 min after consumption of the coated capsule with 240 mL of water. Anatomical landmarks are indicated by numbered labels: heart (**1**), liver (**2**), stomach fundus (**3**), transverse colon (**4**), gluteus medius muscle (**5**) and bladder (**6**). (**A**) The filled capsule appears bright and is located in the ascending colon/hepatic flexure of the large intestine. The capsule here appeared deformed and unevenly filled. The inset (**B**) is from the corresponding fat only MRI image, capturing oil leaks out of the capsule into the colon indicated by the red arrows.

**Figure 4 pharmaceutics-14-00270-f004:**
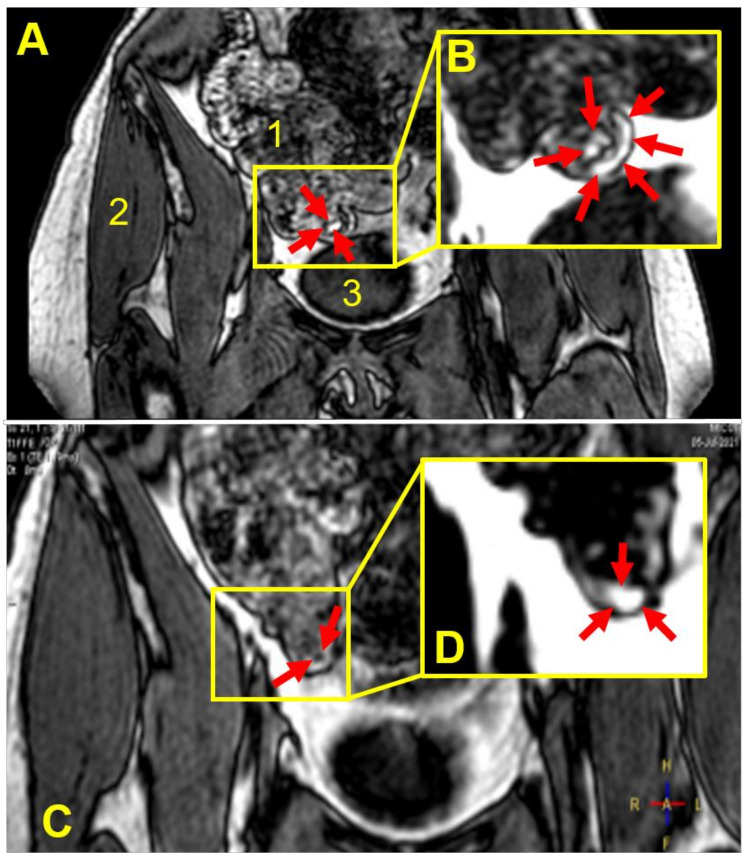
Coronal, fat and water out of phase MRI images of the abdomen of participant number 5 acquired 135 min (**A**) and 180 min (**C**) after consumption of the coated capsule with 240 mL of water. Anatomical landmarks are indicated by numbered labels: ascending colon/caecum (**1**), gluteus medius muscle (**2**) and bladder (**3**). The filled capsule appears bright and is located in the cecum/ascending colon of the large intestine and appeared to be deformed and partially filled (**A**), the capsule being indicated by the red arrows inside the yellow box. The corresponding fat only image (inset (**B**)) captured oil leaks out of the capsule into the colon, indicated by the red arrows. At the later time point (**C**), the capsule, indicated by the red arrows inside the yellow box, appeared to have emptied further and provided some water signal component as well, possibly due to hydration/mixing. The corresponding fat only image (inset (**D**)) captured a cloud of oil filling that leaked out of the capsule into the colon, indicated by the red arrows.

**Table 1 pharmaceutics-14-00270-t001:** Disintegration tests of capsules with different amounts of coating in acid (simulated stomach) and buffer (simulated intestinal) conditions.

		**Acid (Simulated Stomach) Stage** ^**a**^						
	Time to loss of capsule integrity (min)				Time to complete capsule disintegration (min)			
Weight gain (mg)	Capsule 1	Capsule 2	Capsule 3	Mean ± SD	Capsule 1	Capsule 2	Capsule 3	Mean ± SD
0 (uncoated)	5.3	8.7	4.5	6.2 ± 2.2	9.1	14.7	10.7	11.5 ± 2.9
9.2 ± 0.8	Intact	Intact	Intact	-	Intact	Intact	Intact	-
18.2 ± 1.2	Intact	Intact	Intact	-	Intact	Intact	Intact	-
25.9 ± 1.5	Intact	Intact	Intact	-	Intact	Intact	Intact	-
		**Buffer (Simulated Intestinal) Stage ^b^**						
	Time to loss of capsule integrity (min)				Time to complete capsule disintegration (min)			
Weight gain (mg)	Capsule 1	Capsule 2	Capsule 3	Mean ± SD	Capsule 1	Capsule 2	Capsule 3	Mean ± SD
0 (uncoated)	N/A	N/A	N/A	-	N/A	N/A	N/A	-
9.2 ± 0.8	17.8	11.8	16.8	15.5 ± 3.2	>60 ^c^	>60 ^c^	>60 ^c^	-
18.2 ± 1.2	139.3	99.4	112.0	116.9 ± 20.4	>375 ^d^	>375 ^d^	>375 ^d^	-
25.9 ± 1.5	Intact	Intact	Intact	-	-	-	-	-

^a^ 700 mL 0.1M hydrochloric acid at 37.4 °C. ^b^ 700 mL pH = 6.8 phosphate buffer at 37.4 °C. ^c^ The capsule had disintegrated entirely, but some capsule debris was still visible. ^d^ Up to this time, the capsule had shown minor, observable coating changes but without oil release. N/A, capsule had already lost its integrity at the acid (stomach) stage.

**Table 2 pharmaceutics-14-00270-t002:** Summary of the specifications of the capsules administered to the healthy human participants and their location and integrity in the gastrointestinal tract at different time points post ingestion.

Participant	Weight Gain (mg) ^a^	Weight Gain Per Surface Area (mg/mm^2^)	Gastrointestinal Location and Integrity of the Capsule at the Different Imaging Time Points (min)							
			45	90	135	180	225	270	315	360
1	9.2 ± 0.8	0.02	Stomach	Stomach	*Stomach* ^b^	NO ^c^	NO	NO	NO	NO
2	18.2 ± 1.2	0.04	Stomach	Stomach	Duodenum	Duodenum	Term ileum	NO	NO	^d^
3	18.2 ± 1.2	0.04	Stomach	Stomach	Term ileum	*Asc colon*	*Hep flexure*	*Hep flexure*	*Hep flexure*	*Hep flexure*
4	18.2 ± 1.2	0.04	Stomach	Jejunum	Jejunum	Jejunum	*Term ileum*	*Term ileum*	*Term ileum*	NO
5	18.2 ± 1.2	0.04	Stomach	Jejunum	*Cecum*	*Cecum*	*Asc colon*	*Asc colon*	NO	*Hep flexure*
6	18.2 ± 1.2	0.04	Stomach	Stomach	Term ileum	*Term ileum*	NO	NO	NO	NO
7	18.2 ± 1.2	0.04	Stomach	Jejunum	Jejunum	*Term ileum*	*Term ileum*	NO	NO	NO
8	36.0 ± 5.2	0.08	Stomach	Jejunum	Term ileum	*Cecum*	NO	NO		shade
9	52.6 ± 9.7	0.11	Stomach	Duodenum	Term ileum	Hep flexure	Hep flexure	Trans colon	Trans colon	
10	52.6 ± 9.7	0.11	Stomach	Stomach	Duodenum	*Duodenum*	NO	NO		

^a^ Mean ± standard deviation of triplicate measures measured 24 h after coating. ^b^ When loss of integrity of the capsule (deformation of the capsules and/or release of some or all of the oil filling) was detected in the images, this is indicated by italic font. ^c^ NO, Not Observed, indicates when it was not possible to observe the capsule in the MRI images. ^d^ Grayed out cells indicate no imaging was performed at that time point. Abbreviations: Asc, ascending; Hep, hepatic; Term, terminal; Trans, transverse.

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
