# Peer review of "Application of In Vivo MRI Imaging to Track a Coated Capsule and Its Disintegration in the Gastrointestinal Tract in Human Volunteers"

_pharmaceutics, 2022, doi:10.3390/pharmaceutics14020270_

Round 1

Reviewer 1 Report

In this paper, the authors present their work aiming to validate a method to track a coated capsule, and the disintegration of the capsule, through the in vivo GI tract in humans, using MRI. The authors provide an interesting method that is potentially valuable to the field of delayed-release oral drug delivery.

However, the results appear premature, and in my opinion, do not support the claims and conclusions of the authors. The data is very early pilot data, and is substantially limited in scope.

My primary concern is that many of the data points report that the capsule was “not detected”, including 10% (1/10) of the 135 min data points, and 60% (6/10) of the 270 min data points. This includes many of the capsules with the highest coating weight and coating weight gain, suggesting that they should have been detected later in the GI tract. Furthermore, the disintegration of the capsules was only observed in 30% (3/10) of the participants, and in one of those participants, the capsule was observed to be “disintegrating” in caecum at 135 min, and also in the ascending colon at 270 min. How can it be disintegrating at both of these locations and time points? That suggests low precision of this method, and does not support the claims of the authors. Therefore, it appears that disintegration of the capsule was actually only observed in 20% (2/10) participants. It would be beneficial to see a more complete data table, since only limited data is reported in Table 1, which seems to be the critical component of the cohort-level data for this study. The cases presented in Figures 2, 3, and 4 provide nice images and examples ‘case studies’ of this method, but the cohort-level data and trends are insufficiently reported.

Finally, the environment in which these studies were performed was highly controlled with fasting, standard meal timing, etc. This is not representative of real-world medication use, so I am not convinced that the results from this MRI study would actually correlate with the real-world drug delivery data that motivates this study.

In summary, the utility of this method is substantially limited by the inability to trace 60% of the capsules, and the inability of this method to determine the location of disintegration in 80% of the capsules. This is essentially a very early pilot data. The method is indeed potentially valuable, but requires substantial improvement, iteration, and further validation before the authors claims and conclusions would be supported, which are not supported by the current data. I suggest that the authors re-frame this as a pilot / feasibility study.

Author Response

REVIEWER 1

We are grateful to Reviewer 1 for their comments. We have answered all the points raised as detailed below. All changes made to the manuscript can be found in the uploaded ‘tracked changes’ version of the revised manuscript and identified below by the line numbers of the ‘clean’ revised version of the revised document for ease of reference. We have also worked extensively on the text and English language. We believe that following all the changes in response to the reviews the manuscript has substantially improved.

 Comments and Suggestions for Authors

In this paper, the authors present their work aiming to validate a method to track a coated capsule, and the disintegration of the capsule, through the in vivo GI tract in humans, using MRI. The authors provide an interesting method that is potentially valuable to the field of delayed-release oral drug delivery.

However, the results appear premature, and in my opinion, do not support the claims and conclusions of the authors. The data is very early pilot data, and is substantially limited in scope.

Answer:

We agree with the Reviewer’s comment on the potential value of the MRI method described in the manuscript in oral delivery field, and would emphasise that we agree with the reviewer that there are further measurements and studies that one can undertake to take the presented MRI work forward. For this reason, we have performed new in vitro disintegration tests of the uncoated and coated capsules which are now added to the revised manuscript in the new Section 2.2 and Section 2.3. and new Table 1. We will obviously consider further developments of this work stream.

What we are presenting in the manuscript is a feasibility study, with new and exciting data on the proof of applicability of the applied MRI technique in the oral delivery research, whereby a further method development and optimization would make it available for the researchers in the field. We agree that the feasibility nature of this work was not sufficiently emphasised, we have now done this in the revised manuscript abstract Line 21, aims Line 82, design line 128 and Discussion Lines 288 and Conclusions line 387.

  1. My primary concern is that many of the data points report that the capsule was “not detected”, including 10% (1/10) of the 135 min data points, and 60% (6/10) of the 270 min data points. This includes many of the capsules with the highest coating weight and coating weight gain, suggesting that they should have been detected later in the GI tract.

Furthermore, the disintegration of the capsules was only observed in 30% (3/10) of the participants, and in one of those participants, the capsule was observed to be “disintegrating” in caecum at 135 min, and also in the ascending colon at 270 min. How can it be disintegrating at both of these locations and time points? That suggests low precision of this method, and does not support the claims of the authors. Therefore, it appears that disintegration of the capsule was actually only observed in 20% (2/10) participants. It would be beneficial to see a more complete data table, since only limited data is reported in Table 1, which seems to be the critical component of the cohort-level data for this study. The cases presented in Figures 2, 3, and 4 provide nice images and examples ‘case studies’ of this method, but the cohort-level data and trends are insufficiently reported.

Answer:

Thank you so much for raising this point. In answer to your comment we have gone back to all the images, allowing us to generate a new Table 2 which displays all the information available at all time points for all participants, both in terms of capsules’ location and integrity. We hope that the new table satisfies your point about reporting sufficiently cohort-level data and trends.

We agree that the use of the word ‘disintegrating’ was not the most accurate and we have now substituted it with ‘loss of integrity’ meaning that our dosage form is detected as deformed and /or releasing some of the oil content, sometimes a slow release, which is in turn detected at different time points and locations.

In the new Table 2 the loss of integrity is indicated by Italic font for that location, allowing the reader to appraise the entire journey of each capsule.

This process has also allowed us to correct some mistakes in the older text, which we can only apologise for. Loss of integrity was observed directly for 8/10 of the capsules. In one of the studies (participant 2) the capsule was imaged intact at one time point and was entirely no longer visible at the following time points, which is consistent with rapid loss of integrity in between t=225 min and t=270 min. In the last case (Participant 9) the capsule was imaged intact until we could not continue imaging on that day.

We agree that there remains some variability between distance travelled in the gut and level of coating and this is most likely due to individual variability between manual capsule manufacturing and also participants’ physiology.

There is one participant (number 5) for whom at time point at t=315 the capsule was no longer visible but detected again at the following time point. This was most probably due to motion during the imaging procedure, all of which is commented on in Discussion. Lines 364-373 reading: “In Table 2, when the capsule presence in the image stacks could not be detected, this was labeled as ‘Not Observed’. In most cases this has occurred after the capsule was at the previous time point observed to be losing its shape and integrity, or release of encapsulated oil was noticed (text in italic in Table 2), therefore losing the capsule’s imaging signal. In some cases (for example in participant No 5) this might also have meant that at the time of data acquisition physiological motion could have blurred the capsule signal, effectively masking the visual identification of the capsule’s presence. The three-dimensional image acquisition scheme used in this study can be particularly sensitive to motion, and one could in future deploy two-dimensional, multi-slice acquisition sequences to minimize the effect of motion on the images”

  1. Finally, the environment in which these studies were performed was highly controlled with fasting, standard meal timing, etc. This is not representative of real-world medication use, so I am not convinced that the results from this MRI study would actually correlate with the real-world drug delivery data that motivates this study.

Answer:

  1. In summary, the utility of this method is substantially limited by the inability to trace 60% of the capsules, and the inability of this method to determine the location of disintegration in 80% of the capsules. This is essentially a very early pilot data. The method is indeed potentially valuable, but requires substantial improvement, iteration, and further validation before the authors claims and conclusions would be supported, which are not supported by the current data. I suggest that the authors re-frame this as a pilot / feasibility study.

 Answer:

This point has been answered above by the presentation of the complete data in Table 2 and some corrections, showing that tracking of all the capsules and direct imaging of loss of integrity (‘disintegration’) in 8/10 participants. Again, apologies for the initial confusion.

We have nevertheless taken on board the comment about toning down the conclusions and re-framed the paper more clearly as a feasibility study throughout e.g. abstract Line 21, aims Line 82, design line 128 and Discussion Lines 288 and Conclusions line 387

Reviewer 2 Report

In general article is interested in terms of applied MRI techniques for in vivo study. However, some points need to be addressed prior to publication of this manuscript.

  • In coating process, the capsules were coated using dip-coating process, consequently, some areas of the capsule might be coated or uneven coated. How do the authors confirm that the capsules were completely coated? This should be presented in the manuscript.
  • In addition, did the authors measure thickness of the coats on capsule shell?
  • How did the authors measure surface area of capsules?
  • In Table 1, participants received capsules with different coating weight gained. Why did the number of participants who received 18.2 mg coating weight gained capsules are more than the others?
  • From my point of view, “not detected” mean the capsules were completely dissolved in intestinal tract and/or olive oil was disposed from the capsules. What is “not detected” mean?
  • In this study, olive oil was selected for MRI-visible marker fluid. Why did the authors select olive oil? If not olive oil, what are the other markers which could be selected for MRI-marker?

Author Response

REVIEWER 2

We are grateful to Reviewer 2 for their comments. We have answered positively to all the points raised as detailed below. All changes made to the manuscript can be found in the uploaded ‘tracked changes’ version of the revised manuscript and identified below by the line numbers of the ‘clean’ revised version of the revised document for ease of reference.

We have also worked extensively on the text and English language. We believe that following all the changes in response to the reviews the manuscript has substantially improved.

Comments and Suggestions for Authors

In general article is interested in terms of applied MRI techniques for in vivo study.

Answer:

Thank you for confirming interest in the applied MRI techniques in vivo described in our manuscript.

However, some points need to be addressed prior to publication of this manuscript.

In coating process, the capsules were coated using dip-coating process, consequently, some areas of the capsule might be coated or uneven coated. How do the authors confirm that the capsules were completely coated? This should be presented in the manuscript.

Answer:

We agree that the manual coating process has limitations. A thorough visual check was performed following every dipping cycle and after completion, and capsules with visible lack of uniform coating were discarded. This has now been added to the Methods section at Lines 99-102 and the added text reads: “Following the coating procedure, the coated capsules were visually inspected for visible unevenness of the coating and only those capsules with smooth surface appearance were used in the study”

In addition, did the authors measure thickness of the coats on capsule shell?

Answer:

We expressed the coating as mass gain of the capsules. We understand that providing the thickness of the coating layer may be a neater number but that typically used SEM inspection does not confirm the capsule is homogeneously coated. Micro CT may be able to do this however this would be a further study to be undertaken in the future

How did the authors measure surface area of capsules?

Answer:

Apologies for omitting this explanation.  The surface area of a capsule was calculated from known length and diameter using spherocylinder area formula.  This has now been added to the Methods lines 111-112.The added text reads: “The surface area of a capsule was calculated using the spherocylinder area formula, knowing the length and diameter of the capsule.”

In Table 1, participants received capsules with different coating weight gained. Why did the number of participants who received 18.2 mg coating weight gained capsules are more than the others?

Answer:

Thank you, we agree this was not clearly explained. The choice of type of Eudragit polymer used and the concentration in coating solution, was based on literature, some information on weight gain could also be found. The choice of coating was also guided by the experiments, which were not carried out all in one go, and the preliminary data suggested that the 18.2 mg performed well for the remit. This capsule coating was therefore reproduced more frequently, with some studies performed either side with a capsule with about half this amount of coating and some more heavily coated capsules up to 52.6 mg weight gain (we also corrected a weight gain typo at participant 7). We expand now this point in Discussion Lines 329-349. The revised manuscript now says: ‘’ The disintegration test undertaken on Eudragit® S 100 coated HPMC capsules illustrates dependence of the capsules disintegration on weight gain due to coating. Uncoated capsules disintegrated in acidic environment (simulated stomach conditions), whilst all tested coated capsule remained intact for the duration of the acidic environment test (1 hour). In pH 6.8 buffer, simulating intestinal conditions, capsules with 9.2 mg of coating material lost their integrity to release the encapsulated oil, and then disintegrated entirely in less than 20 minutes. On the other hand, the capsules with 18.2 and 25.9 mg of coating layer maintained their integrity in the intestinal buffer stage for approximately 2 and 6 hours respectively, until the release of oil was observed, indicating that the intestinal capsule integrity is proportional to the coating weight gain. These in vitro findings corroborate the in vivo MRI imaging findings. More specifically, the capsule with 9.2 mg weight gain (administered to participant No 1) showed loss of integrity while still present in the stomach. In the participants that were administered capsules with the weight gain of 18.2 and higher, the capsules were emptied from the stomach intact and travelled further down the gastrointestinal tract. The MRI data obtained initially guided iteratively the selection of coating, so that the performance of the capsule with 18.2 mg coating during in vitro as well as in vivo studies meant that these were predominantly used in the study. In three participants we tested capsules with much higher coating gain (36.0 and 52.5 mg) and resistance to disintegration, the latter shown in the in vitro disintegration test, and observed loss of their integrity in two participants, with intact capsule observed in trans-verse colon region of one participant. ‘’.

From my point of view, “not detected” mean the capsules were completely dissolved in intestinal tract and/or olive oil was disposed from the capsules. What is “not detected” mean?

Answer:

We agree that the wording ‘’not detected’’ was not entirely clear and we have now substituted it, in the revised and much more detailed Table 2 with “Not Observed” or NO for short. This usually represent the situation where, as pointed by the reviewer, the capsule had undergone a complete loss integrity and/or complete disposition of the olive oil. It could also mean that at the time of data acquisition physiological motion may have masked the visual identification of the capsule’s presence. There is in Table 1 a good example of this with Participant 5 for whom at time point at t=315 the capsule was no longer visible but detected again at the following time point.  This is now discussed in more detail in Discussion with the added text at Lines 364-373 reading: “I In Table 2, when the capsule presence in the image stacks could not be detected, this was labeled as ‘Not Observed’. In most cases this has occurred after the capsule was at the previous time point observed to be losing its shape and integrity, or release of encapsulated oil was noticed (text in italic in Table 2), therefore losing the capsule’s imaging signal. In some cases (for example in participant No 5) this might also have meant that at the time of data acquisition physiological motion could have blurred the capsule signal, effectively masking the visual identification of the capsule’s presence. The three-dimensional image acquisition scheme used in this study can be particularly sensitive to motion, and one could in future deploy two-dimensional, multi-slice acquisition sequences to minimize the effect of motion on the images.”.

We have also clarified this in Methods Lines 150-154 as follows: ” At each time point, for each participant, the gastrointestinal tract location of the capsule was noted from the image data. Also, the capsule’s intact appearance or loss of integrity, intended as apparent deformation of the capsules and/or release of some or all the oil filling, was noted. When the capsule presence in the image stacks could not be detected, this was labelled NO or Not Observed”.

In this study, olive oil was selected for MRI-visible marker fluid. Why did the authors select olive oil?

Answer:

Olive oil was chosen as a MRI-visible marker fluid for two principal reasons. The first one is that by its inherent nature MRI can image fat and water very well, separately and in different algebraic combinations allowing one to see well the signal from a fat filled capsule against the surrounding organs. We have experience in this kind of imaging and particularly in imaging small, inert fat-containing capsules in the distal bowel to assess gastrointestinal transit, as shown in one of our recent publications for transit markers [1].

The second reason for selecting olive oil is that it is a cheap and safe ‘agent’, which overcomes safety concerns and ethical issues arising from release of the oil in the bowel of the human participants in the study.

We agree this could have been discussed in more detail. We have therefore moved the previous paragraph in Methods (which actually was more discussion matters than methods) into Discussion and added to it, to this effect, in the revised manuscript in Discussion Lines 288-297 and Lines 313-321, which now read: “Olive oil was chosen as a MRI-visible marker fluid filling for the capsules for two principal reasons. The first one is that typically conventional MRI scanners have the unique ability to capture and image signal from fat separately from the water signal and to produce different algebraic combinations of fat and water in phase or out of phase from each other. The olive oil contained in the coated capsules provides high MRI fat signal and virtually no MRI water signal therefore allowing to image the intact capsule as well as oil re-leased from a capsule that has lost its integrity. Here, as illustrated in the MRI images in Figures 2-4, the fat and water out of phase imaging combination was able to clearly visualize the capsule, whilst providing good anatomical detail, enabling assessment of the capsule’s spatial location within the gastrointestinal tract. Furthermore, considering imaging of oil-containing formulations in the distal intestinal tract, one would not expect to find much fluid fat in the chyme in the colon and therefore the signal from the oil in the capsule should be clearly visible. The second reason is that olive oil is an inexpensive, safe, food-grade material, and there are no ethical issues arising from the capsules disintegrating and releasing oil in the bowel of healthy human participants”.

If not olive oil, what are the other markers which could be selected for MRI-marker?

Answer:

Positive MRI contrast from paramagnetic agents such as gadolinium (Gd) chelates have been used in the past by us and others though recently safety concerns have been raised [2, 3]. Other natural oral MRI contrast materials are pineapple juice or pieces, date syrup, milk and fruit juice [4-7]. However all these agents modulate the properties of the water signal only, whilst our approach exploits as described above the fat signal too thereby providing wider opportunities for imaging and detection.

  1. Sharif, H.; Abrehart, N.; Hoad, C.; Murray, K.; Perkins, A.; Smith, M.; Gowland, P.; Spiller, R.; Harris, R.; Kirkham, S.; et al. Feasibility Study of a New Magnetic Resonance Imaging Mini-capsule Device to Measure Whole Gut Transit Time in Paediatric Constipation. J Pediatr Gastroenterol Nutr 71: 604-611, 2020, doi:10.1097/MPG.0000000000002910.
  2. Layne, K.A.; Dargan, P.I.; Archer, J.R.H.; Wood, D.M. Gadolinium deposition and the potential for toxicological sequelae - A literature review of issues surrounding gadolinium-based contrast agents. Br. J. Clin. Pharmacol. 2018, 84, 2522-2534, doi:10.1111/bcp.13718.
  3. Runge, V.M. Dechelation (Transmetalation) Consequences and Safety Concerns With the Linear Gadolinium-Based Contrast Agents, In View of Recent Health Care Rulings by the EMA (Europe), FDA (United States), and PMDA (Japan). Invest. Radiol. 2018, 53, 571-578, doi:10.1097/rli.0000000000000507.
  4. Arthurs, O.J.; Graves, M.J.; Edwards, A.D.; Joubert, I.; Set, P.A.; Lomas, D.J. Interactive neonatal gastrointestinal magnetic resonance imaging using fruit juice as an oral contrast media. BMC medical imaging 2014, 14, 33, doi:10.1186/1471-2342-14-33.
  5. Govindarajan, A.; Lakshmanan, P.M.; Sarawagi, R.; Prabhakaran, V. Evaluation of Date Syrup as an Oral Negative Contrast Agent for MRCP. American Journal of Roentgenology 2014, 203, 1001-1005, doi:10.2214/AJR.13.12299.
  6. Elsayed, N.M.; Alsalem, S.A.; Almugbel, S.A.A.; Alsuhaimi, M.M. Effectiveness of natural oral contrast agents in magnetic resonance imaging of the bowel. The Egyptian Journal of Radiology and Nuclear Medicine 2015, 46, 287-292, doi:https://doi.org/10.1016/j.ejrnm.2015.03.007.

Grimm, M.; Ball, K.; Scholz, E.; Schneider, F.; Sivert, A.; Benameur, H.; Kromrey, M.L.; Kühn, J.P.; Weitschies, W. Characterization of the gastrointestinal transit and disintegration behavior of floating and sinking acid-resistant capsules using a novel MRI labeling technique. Eur J Pharm Sci 2019, 129, 163-172, doi:10.1016/j.ejps.2019.01.012.

Reviewer 3 Report

In this paper, the authors report in vivo behavior of coated capsules with intestinosolvens Eudragit. The research includes a human experiment that shows exciting and interesting results, but there are several weaknesses.

  1. The static measurements are missing from the test of capsule weight gain measure. In a high concentration of Eudragit solution, the capsule weight gain is not significantly different after the dip-coating cycle. To make statistics, it would be necessary to increase the number of replicates to n=20.
  2. A study of the formed coating is missing from the manuscript, an increase in the weight of the capsules does not mean that the coating is properly formed. Image formation methods such as SEM (Scanning electron microscope), micro-CT should be used to examine the completeness and perfection of the coating.
  3. The in vivo results show high standard deviation, which is not a problem, but their interpretation is difficult and uncertain without in vitro The gold standard to prove gastric acid resistant coating is European Pharmacopoeia 10.0, Dissolution test dissolution test for solid dosage forms (chapter 2.9.3.), delayed-release solid dosage forms. This must be done before publication. 

https://dl1.cuni.cz/pluginfile.php/1089709/mod_resource/content/2/dissolution%20testing.pdf

It is advisable to test the disintegration of the uncoated capsule also during the experiment, because depending on the type of HPMC capsules may delay disintegration on their own.

https://www.capsugel.com/consumer-health-nutrition-products/drcaps-capsules

The paper is not acceptable for publication in Pharmaceutics before the above studies are performed.

Author Response

REVIEWER 3

We thank the Reviewer for their overall appraisal that the results of the work presented are “exciting and interesting”. We provide a detailed answer to the points raised below. All changes made to the manuscript can be found in the uploaded ‘tracked changes’ version of the revised manuscript and identified below by the line numbers of the ‘clean’ revised version of the revised document for ease of reference. We have also worked extensively on the text and English language. We believe that following all the changes in response to the reviews the manuscript has substantially improved.

Comments and Suggestions for Authors

In this paper, the authors report in vivo behavior of coated capsules with intestinosolvens Eudragit. The research includes a human experiment that shows exciting and interesting results, but there are several weaknesses.

  1. The static measurements are missing from the test of capsule weight gain measure. In a high concentration of Eudragit solution, the capsule weight gain is not significantly different after the dip-coating cycle. To make statistics, it would be necessary to increase the number of replicates to n=20.

Answer:

We agree with the reviewer that we should have added standard deviation data. Apologies for the omission. This is now added to the Table 1 in the revised manuscript (based on n=3 including the standard deviation).  Data on 4 different amounts of coating are presented in Table 1. We agree that in the case of low Eudragit concentrations used for coating the weight gain is small (in a few mg range). One would expect that as small amount of Eudragit material is present in the solution for the coating that the weight gain sample is small and that it would not allow us to draw strong statistical inferences between coating cycles and Eudragit solution concentrations. Indeed we had not attempted to make statistical inferences. Due to the feasibility nature of the study we were not able to make a substantial number of replicates.

  1. A study of the formed coating is missing from the manuscript, an increase in the weight of the capsules does not mean that the coating is properly formed. Image formation methods such as SEM (Scanning electron microscope), micro-CT should be used to examine the completeness and perfection of the coating.

Answer:

We agree with the reviewer’s comment that further analysis, in addition to weight gain, of the coating layer should be undertaken in the future. We understand that calculated mass gain/surface area assumes formation of a homogeneous surface layer which in practice may not be the case. However suggested SEM measurements would provide a valuable information on the layer thickness at certain measured points at the capsule surface, but would not provide a confirmation that that layer is homogeneous throughout the surface, i.e. completeness and perfection of coating.

We appreciate the point and have added the following text to the revised manuscript at Methods Lines 99-102 “Following the coating procedure, the coated capsules were visually inspected for visible unevenness of the coating and only those capsules with smooth surface appearance were used in the study and their data on mass gain measurements reported..” and also added this text to Discussion Lines 353-356 “We have visually inspected all capsules tested and used in the MRI study for any unevenness and selected only those with even, smooth coating. In further studies and optimizations, one would aim to measure the layer thickness and its uniformity at the capsule shell”.

  1. The in vivo results show high standard deviation, which is not a problem, but their interpretation is difficult and uncertain without in vitro The gold standard to prove gastric acid resistant coating is European Pharmacopoeia 10.0, Dissolution test dissolution test for solid dosage forms (chapter 2.9.3.), delayed-release solid dosage forms. This must be done before publication. 

https://dl1.cuni.cz/pluginfile.php/1089709/mod_resource/content/2/dissolution%20testing.pdf

It is advisable to test the disintegration of the uncoated capsule also during the experiment, because depending on the type of HPMC capsules may delay disintegration on their own.

https://www.capsugel.com/consumer-health-nutrition-products/drcaps-capsules The paper is not acceptable for publication in Pharmaceutics before the above studies are performed.

Answer:

Thank you for raising this important point. In direct response we have now teamed up with a laboratory specialist in our Pharmacy department, Matthew Calladine (who has now been added also to the co-authors of the manuscript in light of this contribution) and performed disintegration tests using a USP compatible basket apparatus and stomach and intestinal conditions as per the US Pharmacopoeia dissolution tests. We tested uncoated capsules as suggested, and three different levels of coating.

The tests confirmed our predictions on acid resistance and delayed disintegration time in intestinal conditions. We have accordingly added a new Section 2.2 in Methods (with 2 new references) at Lines 115-124 which reads: “The disintegration behavior of the uncoated and coated capsules was tested in the USP compatible basket apparatus (ZT 120 light, ERWEKA GmbH, Langen, Germany). In this protocol 700 mL of 0.1 M hydrochloric acid was used for the ‘acid stage’, simulating the gastric fluid, and the capsules were exposed for 1 hour. The capsules were then immersed into 700 mL of phosphate buffer pH 6.8, termed here the ‘buffer stage’, simulating the intestinal fluids. The experiments were performed at 37 ± 2 ⁰C and a stirring speed of 30 ± 1 strokes/min as per the United States Pharmacopeia (USP) [34,35]. Triplicates of un-coated and capsules coated by 1, 2 or 3 dipping cycles in 10.7% Eudragit® S 100 coating solution were tested. Time to loss of capsules integrity, as well as to complete disintegration of capsules were recorded”.

A new Table 1 and a new Section 3.2 was also added in the Results section at Lines 184-194 as follows: “The data from the disintegration tests in acid (stomach) and buffer (intestinal) conditions are reported in Table 1. In the disintegration tests (in triplicates) the uncoated capsules lost their integrity and released their first drop of oil in 6.2 ± 2.2 min in the simulated stomach (acid) stage. The uncoated capsules had disintegrated entirely within 11.5 ± 2.9 min in the acid stage. All the coated capsules remained intact in the acid stage. The capsules manufactured using the lowest amount of coating (9.2 mg) released their first drop of oil in 15.5 ± 3.2 min during the intestinal (buffer) stage. The capsules manufactured using 18.2 mg of coating lost their shape and released the first drop of oil after 116.9 ± 20.4 min in the buffer stage whereas the capsules manufactured using the higher 25.9 mg coating maintained their integrity in the buffer stage for over 6 hours, when the disintegration experiment was stopped, with no signs of loss of shape or oil release.”.

Finally, a paragraph was added to Discussion at Lines 322-349 as follows:

“The design of the gastroresistant, polymer coated capsule used in this study, the choice of specific Eudragit® polymer and the range of concentrations of the coating solution was based on the literature [37-51]. The selection of HPMC as the capsule shell took into consideration that HPMC is often used as a pre-coating material in gastroresistant / enteric coated formulations as gelatin capsules do not appear to have good adhesion properties when directly coated with Eudragit® polymers [39,48]. In addition, HPMC properties seem to favor the gradual drug release in the colon [44].

The disintegration test undertaken on Eudragit® S 100 coated HPMC capsules illustrates dependence of the capsules disintegration on weight gain due to coating. Uncoated capsules disintegrated in acidic environment (simulated stomach conditions), whilst all tested coated capsule remained intact for the duration of the acidic environment test (1 hour). In pH 6.8 buffer, simulating intestinal conditions, capsules with 9.2 mg of coating material lost their integrity to release the encapsulated oil, and then disintegrated entirely in less than 20 minutes. On the other hand, the capsules with 18.2 and 25.9 mg of coating layer maintained their integrity in the intestinal buffer stage for approximately 2 and 6 hours respectively, until the release of oil was observed, indicating that the intestinal capsule integrity is proportional to the coating weight gain. These in vitro findings corroborate the in vivo MRI imaging findings. More specifically, the capsule with 9.2 mg weight gain (administered to participant No 1) showed loss of integrity while still present in the stomach. In the participants that were administered capsules with the weight gain of 18.2 and higher, the capsules were emptied from the stomach intact and travelled further down the gastrointestinal tract. The MRI data obtained initially guided iteratively the se-lection of coating, so that the performance of the capsule with 18.2 mg coating during in vitro as well as in vivo studies meant that these were predominantly used in the study. In three participants we tested capsules with much higher coating gain (36.0 and 52.5 mg) and resistance to disintegration, the latter shown in the in vitro disintegration test, and observed loss of their integrity in two participants, with intact capsule observed in trans-verse colon region of one participant.”.

Reviewer 4 Report

The manuscript entitled "Application of in vivo MRI imaging to track a coated capsule and its disintegration in the gastrointestinal tract in human volunteers" outlines the MRI-based methodology to monitor the disintegration of an HPMC/Eudragit capsule containing olive oil along the gastrointestinal tract in order to evaluate an alternative method to ionising radiation or commonly used contrast agents.

The article was well written and presented relevant information on capsule preparation, imaging and explanation of capsule disintegrations along the gastrointestinal tract. In my opinion, it would be interesting to develop further studies on the bioavailability of olive oil bioactive compounds entrapped in HPMC/Eudragit capsules to assess possible interactions between the coatings and the bioactive compounds. Despite the good information provided by the authors, I consider that minor revisions need to be made as some missing information should be included.

Line 81: It would be useful to add the standard deviation of the weight gained in Table 1, as the production of the capsules is very manual, it would be interesting to give readers an objective view of the reproducibility of the process.

After the authors have incorporated these data, I consider this work ready for publication.

Author Response

REVIEWER 4

We are grateful to Reviewer 4 for the positive appraisal of our manuscript. We have answered positively to all the points raised as detailed below. All changes made to the manuscript can be found in the uploaded ‘tracked changes’ version of the revised manuscript and identified below by the line numbers of the ‘clean’ revised version of the revised document for ease of reference. We have also worked extensively on the text and English language.  We believe that following all the changes in response to the reviews the manuscript has substantially improved.

Comments and Suggestions for Authors

The manuscript entitled "Application of in vivo MRI imaging to track a coated capsule and its disintegration in the gastrointestinal tract in human volunteers" outlines the MRI-based methodology to monitor the disintegration of an HPMC/Eudragit capsule containing olive oil along the gastrointestinal tract in order to evaluate an alternative method to ionising radiation or commonly used contrast agents.

The article was well written and presented relevant information on capsule preparation, imaging and explanation of capsule disintegrations along the gastrointestinal tract. In my opinion, it would be interesting to develop further studies on the bioavailability of olive oil bioactive compounds entrapped in HPMC/Eudragit capsules to assess possible interactions between the coatings and the bioactive compounds.

Answer:

We are unsure what the Reviewer means with olive oil bioactive compounds, we interpreted this as API

We have added a comment to this effect at Lines 356-357 which reads “Furthermore, future studies could evaluate the bioavailability of active pharmaceutical compounds incorporated within the capsule.”.

Despite the good information provided by the authors, I consider that minor revisions need to be made as some missing information should be included.

Line 81: It would be useful to add the standard deviation of the weight gained in Table 1, as the production of the capsules is very manual, it would be interesting to give readers an objective view of the reproducibility of the process.

 Answer:

Agreed, we have now revised extensively Table 2, and also added to it, as suggested, the standard deviations of weight gain with linked footnote.

After the authors have incorporated these data, I consider this work ready for publication.

 Answer:

Thank you again for the positive comments.

Round 2

Reviewer 3 Report

It contains some text editing errors, that should be correct before accepting the manuscript. 

Author Response

Dear Reviewer 3

Thank you so much for these final comments. In response we have removed the graphical abstract from the main Word document, and consequently tidied up the layout of the main text paragraphs. We have sorted out the page numbering and the page headers/footers. We have added the doi to some of the references where this was missing but available online. We have done all this after having accepted all previous tracked changes as there were so many that it was almost impossible to work out the final layout of the text. We have run once more a spell check.

We hope that the above has satisfied the corrections needed for the text editing imperfections. Thank you again for all the help with our manuscript